# Thymidylate synthase promotes esophageal squamous cell carcinoma growth by relieving oxidative stress through activating nuclear factor erythroid 2-related factor 2 expression

Jian Yang[1,2☯]*, Jingjing Zhang[3☯], Jingtian Chen[4], Xiaolong Yang[2], Hui Sun[5], Zhenxiang Zhao[1], Hui Zhou[1], Hao Shen[6]

1 Translational Medicine Research Center, Shanxi Medical University, Taiyuan, Shanxi, PR China, 2 Department of Cell Biology and Genetics, College of Basic Medicine, Shanxi Medical University, Taiyuan, Shanxi, PR China, 3 Department of Physiology, College of Basic Medicine, Shanxi Medical University, Taiyuan, Shanxi, PR China, 4 Department of Colorectal Surgery, The Fifth Clinical Medical College of Shanxi Medical University, Taiyuan, Shanxi, PR China, 5 Science & Technology Information and Strategy Research Center of Shanxi, Taiyuan, Shanxi, PR China, 6 College of Pharmacy, Shanxi Medical University, Taiyuan, Shanxi, PR China

☯ These authors contributed equally to this work.
* jiany@sxmu.edu.cn

**Data Availability Statement:** All relevant data are within the paper and its Supporting Information files.

## Abstract

### Background

Thymidylate synthase (TYMS) is involved in the malignant process of multiple cancers, and has gained much attention as a cancer treatment target. However, the mechanism in carcinogenesis of esophageal squamous cell cancer (ESCC) is little reported. The present study was to clear the biological roles and carcinogenic mechanism of TYMS in ESCC, and explored the possibility to use TYMS as a tumor marker in diagnosis and a drug target for the treatment of ESCC.

### Methods

Stably TYMS-overexpression cells established by lentivirus transduction were used for the analysis of cell proliferation. RNA sequencing was performed to explore the possible carcinogenic mechanisms.

### Results

GEPIA databases analysis showed that TYMS expression in esophageal cancer tissues was higher than that in normal tissues. The MTT assay, colony formation assay, and nude mouse subcutaneous tumor model found that the overexpression of TYMS increased cell proliferation. Transcriptome sequencing analysis revealed that the promoted cell proliferation in TYMS-overexpression ESCC cells were mediated through activating genes expression of nuclear factor erythroid 2-related factor 2 (Nrf2) and Nrf2 dependent antioxidant enzymes to relieve oxidative stress, which was confirmed by increased glutathione (GSH), glutathione peroxidase (GPX) activities, and reduced reactive oxygen species. Nrf2 active

**Funding:** This research was supported by the National Natural Science Foundation of China (82103175), Natural Science Foundation of Shanxi Province (20210302123316).

**Competing interests:** The authors have declared that no competing interests exist.

inhibitors (ML385) used in TYMS-overexpression cells inhibited the expression of Nrf2-dependent antioxidant enzyme genes, thereby increasing oxidative stress and blocking cell proliferation.

## Conclusion

Our study indicated a novel and effective regulatory capacity of TYMS in the cell proliferation of ESCC by relieving oxidative stress through activating expression of Nrf2 and Nrf2-dependent antioxidant enzymes genes. These properties make TYMS and Nrf2 as appealing targets for ESCC clinical chemotherapy.

## Introduction

Esophageal cancer is one of the malignant tumors with high morbidity and mortality all over the world, and the histological types of esophageal cancer are mainly esophageal adenocarcinoma (ECA) and esophageal squamous cell carcinoma (ESCC) [1]. ESCC is a major type of esophageal cancer in China, accounting for 85%- 90% of all esophageal cancer cases [2]. Most ESCC patients are diagnosed at an advanced stage because of the imperceptible clinical symptoms in early stage [3]. The lack of early diagnostic markers, prognostic indicators and specific treatment targets resulted in a low survival rate and high recurrence rate for ESCC patients [4]. Therefore, it is imperative to better understand carcinogenic mechanism of ESCC to explore potential tumor markers and therapeutic targets for the diagnosis and treatment of ESCC [5].

Aerobic glycolysis, known as Warburg effect, supplies most tumor energy in a faster manner by oxidizing glucose rather than oxidative phosphorylation [6], which would also reprogram nucleic acid metabolism. Therefore, more attention was payed to the novel therapeutic strategies for cancer chemotherapy based on the Warburg effect, such as targeting oncogenes and anti-oncogenes to switch the metabolic mode cancer cells. The changed biochemical characterizations in cancer cell energy and material metabolism were modulated by a multi-gene system, providing novel therapeutic strategies for targeted cancer therapies. Targeting nucleic acid metabolism to inhibit tumor cell proliferation is an accurate, effective and promising therapeutic strategy in the clinical treatment of cancer [7]. The metabolic drugs, especially those targeting nucleotide-metabolizing enzymes such as dihydroorotate dehydrogenase and ribonucleotide reductase, have been approved for clinical use, and significantly improved the survival rate of cancer patients. However, tumor targeted therapy is still facing challenges such as drug resistance and serious side effects, and it is warranted to clear the function and mechanisms of key metabolic enzymes in tumor. Our previous research found the amplification frequency of pyrimidine metabolic pathway related genes in ESCC samples was significantly higher than that in paracancerous samples [8], and thymidylate synthase (TYMS) was one of many abnormal genes. TYMS is a key enzyme involved in the enzymatic conversion of methylates deoxyridine monophosphate (dUMP) to deoxythymidine monophosphate (dTMP) [9], and acts as the sole source of thymidine for DNA replication. Xie et al. [10] found the transcription regulation function of TYMS, and monomer and dimer of TYMS play an important role in TYMS mRNA expression regulation, and TYMS without ligands combines its own mRNA to inhibit translation and protein synthesis [11]. The reduced TYMS negatively affected the balance of nucleotides in cells, resulting in the rapid depletion of dTMP and the reduction of dTTP synthesis [12]. TYMS is expressed in almost all tumor cells, and more TYMS is found in non-small cell lung cancer [13], colorectal cancer [14], breast cancer [15],

and cholangiocarcinoma [16]. TYMS is used as the target of anti-tumor drugs 5-fluorouracil to inhibit tumor cells proliferation. Although chemotherapy based on 5-fluorouracil has become an important approach in the treatment of multiple types of tumors, insensitivity to 5-fluorouracil and fatal gastrointestinal adverse reactions were observed in some cases. The congenital or acquired drug resistance of tumor cells resulted that the effective rate was only about 20% for cancer patients with 5-fluorouracil-based treatment, seriously restricting the application of fluorouracil drugs in the cancer clinical treatment [17]. Therefore, it is necessary to propose new treatment schemes to increase the sensitivity of tumor cells to 5-fluorouracil drugs and enhance its anti-tumor effect. This study aims to detect the biological function of TYMS in ESCC and explore its possible molecular mechanisms, so as to provide certain reference value for the selection of therapeutic targets for ESCC.

## Materials and methods

### Ethics statement

This study was carried out in strict accordance with the recommendations in the Guide for the Care and Use of Laboratory Animals of the National Institutes of Health. The protocol was approved by the Committee on the Ethics of Animal Experiments of Shanxi Medical University (Protocol Number: 2021–101). All surgery was performed under sodium pentobarbital anesthesia, and all efforts were made to minimize suffering.

### TYMS mRNA expression analysis in esophageal cancer

Based on the data from Genotype-Tissue Expression projects (GTEx) and Cancer Genome Atlas (TCGA), TYMS mRNA expression in esophageal cancer patients were obtained through the Gene Expression Profiling Interactive Analysis (GEPIA).

### Cell culture, TYMS-overexpression and knockdown of ESCC cells

Human ESCC cell lines used in the present study were obtained from the Key Laboratory of Cellular Physiology and Translational Medical Research Center, Shanxi Medical University (Taiyuan, Shanxi, China). Cells were cultured in RMPI 1640 medium containing 10% calf serum in 5% $CO_2$ at 37°C. To overexpress the TYMS, the specific sequence of TYMS was cloned into the Ubi-MCS-flag-CMV-GFP-IRES-Puro lentivirus expression vector (Shanghai GeneChem Co., Ltd.), and the lentivirus supernatant was used to transfect the human ESCC cell lines KYSE150 and KYSE180. The transfection with TYMS cDNA constructs produced the predicted protein (named as TYMS), and the matching empty vectors were named NC. Slow virus vector sequence 5′-CAACCCTGACGACAGAAGA-3′ (named shTYMS) and 5′-TT CTCCGAACGTGTCACGT-3′ (named shcon) was used for the knockdown of endogenous TYMS expression. The efficiencies of overexpression and knockdown were determined via qRT-PCR and Western blot.

### Cell proliferation analyses

Through MTT assay and soft-agar colony forming assay, cell proliferation was examined. MTT assay: The cells were seeded in 96 well plates at a density of 4000 cells per well. After 24 h, 48 h, 72 h, 96 h and 120 h, 20 μl MTT (5 mg/ml in DMEM) was added to each well. After 4 hours of incubation, 100 ul dimethyl sulphoxide (DMSO) was added. After 10 minutes, the absorbance at 490 nm of each well was measured by Multimode Reader (Varioskan Flash, Thermo Electron Co. US). Soft agar colony forming assay: Each agarose disk was inoculated with 1000 cells, and amended with cell culture medium. After 12 days, cells were fixed for

30min with 5% trioxane, dyed with crystal violet, washed with PBS, and then the number of colony forming was counted.

## Animal experiments

To analyze the effects of TYMS on tumorigenesis *in vivo*, a mouse xenografted assay was carried out. The female BALB/c nude mice were purchased from GemPharmatech (Strain NO. D000521, Nanjing, China). $5 \times 10^6$ TYMS-overexpression KYSE150 cells were subcutaneously injected into 4-week-old female BALB/c nude mice, and KYSE150 cells were used as a control. Tumor size was measured every 7 days using caliper and calculated using the following formula: tumor volume $(mm^3) = 1/2 \times$ length $(mm) \times$ [width $(mm)]^2$. After 5 weeks, mice were executed by anesthesia and tumors were removed and measured. Mice experiments were carried out according to the Laboratory Animal Guide of Shanxi Medical University.

## Western blotting

Cultured cells were lysed in RIPA buffer with protease inhibitor mixtures at 4˚C, and were centrifuged at 12,000 rpm at 4˚C for 30 min. Equal amounts of proteins were separated by 10% SDS-PAGE, and were then transferred onto polyvinylidene fluoride membranes (Whatman, Maidstone, Kent, UK). The separated proteins were probed with the special antibodies including anti-TYMS antibody (15047-1-AP, Proteintech) by incubation overnight at 4˚C. After washing with TBST, the membranes were incubated with HRP-conjugated Affinipure Goat Anti-Robit (SA00001-2, Proteintech) IgG (H+L) secondary antibody for 2 hours. The membranes were finally washed with TBST buffer before the chemiluminescent substrates of western blot were applied with electrochemiluminescence kit (SW133-01, Seven Biotech, Beijing, China), and were detected with Imaging System (Tanon 4600SF, Shanghai, China). The β-actin (81115-1-RR, Proteintech) was used as loading control.

## Transcriptome survey and expression analysis

Total RNA samples were isolated according to the protocols of Trizol (Takara Shuzo Co. Ltd., Kyoto, Japan), and the sequencing library was performed using Illumina hiseq 2500 sequencing platform. The resulting clean reads were *de novo* assembled using Trinity software, the differential expressed genes (DEGs) were calculated through reads per KB per million reads (RPKM) method with false discovery rate $P < 0.05$ and multiple difference $\geq 2$. Gene Ontology (GO) and the Kyoto Encyclopedia of Genes and Genomes pathway database (KEGG) pathway enrichment analysis were carried out to further clear the biological functions of DEGs using the KOBAS2.0 and Blast2GO programs, respectively.

## qRT-PCR analysis

Total RNA was extracted from about $5.0 \times 10^5$ cells using the RNA extraction kit according to the manufacturer's instructions. The concentration of isolated RNA extracted were quantified at 230nm, 260nm, and 280nm using a NanoDrop Spectrophotometer (Thermo Scientific, USA), and only those samples with a 260nm /280nm ratio between 1.8–2.1 and a 260nm/ 230nm ratio of more than 2.0 were used for further analysis. Using MMLV reverse transcriptase (Takara Shuzo Co. Ltd., Kyoto, Japan), the first-strand cDNA was synthesized from the 200 ng of isolated total RNA. Then, gene expression was studied by determining mRNA applying the SYBR® Green PCR Master Mix (Vazyme Biotech Co., Ltd) by the ABI 7500 Real-Time PCR System (Applied Biosystems, Foster City, CA, USA). The qRT-PCR specific primers were designed based on gene sequences from the NCBI GenBank database using Primer

software (Premier Biosoft, Palo Alto, CA, USA), shown in Table 1. The qRT-PCR reaction mixture was shown below: 0.5 μL each forward and reverse primers (10 μM), 1 μL cDNA template, 10 μL Realtime PCR Super mix (2×), and RNase free dH$_2$O to adjust to 20 μL. Using the $2^{-\Delta\Delta Ct}$ method, the relative quantification of mRNA expression was achieved by concurrent amplification of the GAPDH endogenous control.

## Detection of oxidative stress, glutathione, and glutathione peroxidases (GPX) in ESCC cell lines

Based on chemical fluorescence method, reactive oxygen species (ROS) were detected with DCFH-DA (2,7-Dichlorofuorescin diacetate) fluorescent probe. According to the protocol of ROS assay kit (E004-1-1, Nanjing Jiancheng Bioengineering Institute, Nanjing, China), 1 x 10$^6$ cells were added to DCFH-DA (working concentration: 10 μM) in darkness, and were incubated for 15 min at 37°C. Then, these cells were centrifuged at 1, 000 g for 10 min, and were suspended with cold PBS buffer used for the fluorescence detection through flow cytometry.

Following the method of Anderson (1976), reduced glutathione (GSH) and oxidized (GSSG) contents were measured by a commercially available kit (Nanjing Jiancheng Bioengineering Institute, Nanjing, China). At first, total glutathione was determined in the homogenates spectrophotometrically at 412 nm, after precipitation with 0.1 M HCl, using GR, 5, 5'-dithio-bis- (2-nitrobenzoic acid) (DTNB) and NADPH. The levels of GSH and GSSG were expressed as nmol/mg protein.

Collected cells added to sample homogenate were ruptured by ultrasonic treatment, and were then centrifuged at 4°C for 10 min at 12000 rpm. The supernatant was used for the determination of protein concentration and enzyme activity. According to the protocol of glutathione peroxidases (GPX) kit (S0056, Beyotime Biotechnology, China), 2.5μl supernatant were added to 89μl GPX detection buffer, 2.5μl GSH solution (10mM), 6μl peroxide reagent solution (15mM), incubated at 25°C for 10 minutes. Then, added 3.3μl DTNB solution was added to each well, and incubated at 25°C for 10 minutes. The absorbance at 412 nm of each well was measured, and the relative enzyme activity was calculated.

## ML385 treatment

ML385 was purchased from Abmole (M8692, Houston, TX, USA). Stock solutions of 5mM ML385 were prepared in DMSO. All dilutions to working solution were performed in

**Table 1. Specific primer pairs for the TYMS, Nrf2, GCLC, SOD1, SLC7A11 and GAPDH genes used in quantitative real-time polymerase chain reaction analysis.**

| Gene | Primer (5'-3') |
|---|---|
| TYMS | AACCCTGACGACAGAAGA |
| | CGATGTTGAAAGGCACA |
| Nrf2 | GCGACGGAAAGAGTATGAGC |
| | GGGAGTAGTTGGCAGATCCA |
| GCLC | ACGGAGGAACAATGTCCGAG |
| | TGTGAACCCAGGACAGCCTA |
| SOD1 | GAAGGTGTGGGGAAGCATTA |
| | ACATTGCCCAAGTCTCCAAC |
| SLC7A11 | TGCCCAGATATGCATCGTCC |
| | CTTCTTCTGGTACAACTTCCAGT |
| GAPDH | GGAGCGAGATCCCTCCAAAAT |
| | GGCTGTTGTCATACTTCTCATGG |

appropriate cell culture media. In all experiments of measuring the intracellular ROS, glutathi-one, GPX, and genes expression, cells were treated with 5.0μM ML385 for 72h.

## Statistics analysis

Data were presented with mean ± standard deviation (S.D.) from the least three independent experiments. All statistical analysis was performed using SPSS software (version 17.0, SPSS Inc., Chicago, IL, USA), and data graphs were generated using GraphPad Prism 9 Software (GraphPad Software Inc. La Jolla, CA, USA). Data homogeneity of variance and normality were tested using Levene tests and Kolmogorov–Smirnov tests, respectively. $p$ values were performed using Student's t-test and one-way ANOVA analysis between groups, and a value of $p < 0.05$ was considered as statistically significant which was denoted with *.

# Results

## TYMS promotes ESCC cell proliferation and tumor growth

An elevation of TYMS mRNA in esophageal cancer tissues was observed based on the analysis of GEPIA online database (S1 Fig). To study the biological function of TYMS in ESCC, the levels of TYMS mRNA in different ESCC cell lines were at first detected through qRT-PCR and Western blot (Fig 1A). The KYSE150 and KYSE180 cell lines were chosen for a TYMS overexpression experiment, and TE-1 and KYSE180 cell lines were chosen for a TYMS-knockdown experiment. The efficiency in TYMS overexpression and knockdown cells were also confirmed by qRT-PCR and Western blot (Figs 1B and S2A). MTT assay and colony formation assay showed that the overexpression of TYMS significantly promoted the proliferation in both KYSE150 and KYSE180 cells (Fig 1C–1F). Moreover, cell proliferation was inhibited in TYMS-konckdown cells (S2B and S2C Fig). The effect of TYMS on tumorigenesis *in viv*o was evaluated by xenograft mouse model, and the tumor volumes and weight in TYMS-overexpression cells were significantly increased compared to that in the control group (Fig 1G and 1H). These results suggest that TYMS could act as a cancer-promoting gene to promote cell proliferation in ESCC.

## Transcriptomics reveals that TYMS could relieve oxidative stress through activating Nrf2 expression

In order to further explore the regulatory mechanism of TYMS in ESCC, the TYMS-overexpression cells and negative control cells were sequenced by transcriptome. The sequencing results showed that the genes expressions of Nrf2 and Nrf2-dependent antioxidant enzymes such as glutamate- cysteine ligase subunit catalytic (GCLC), copper/zinc superoxide dismutase 1 (SOD1), solute carrier family 7 member 11 (SLC7A11) were significantly up-regulated. This was confirmed by RT-qPCR analysis in TYMS-overexpression and knockdown cells (Fig 2A and 2B and S2D and S2E Fig). Further tests found the significantly increased GSH levels (Fig 2C) and GPX activities (Fig 2D) in TYMS-overexpression cells, and the opposite results could be observed in TYMS-knockdown cells (S2F–S2H Fig). This indicates that more TYMS could enhance intracellular antioxidant capacity to clear ROS (Fig 2E).

## Blocking Nrf2-mediated defense response inhibits TYMS-overexpression cells

Carcinogenesis of TYMS may work by activating Nrf2 expression in human ESCC. To further clarify the regulatory mechanism, a inhibitor (ML385) of Nrf2 activity was used in TYMS-overexpression cells. Results showed that the expression of GCLC, SOD1 and SLC7A11 in ML385

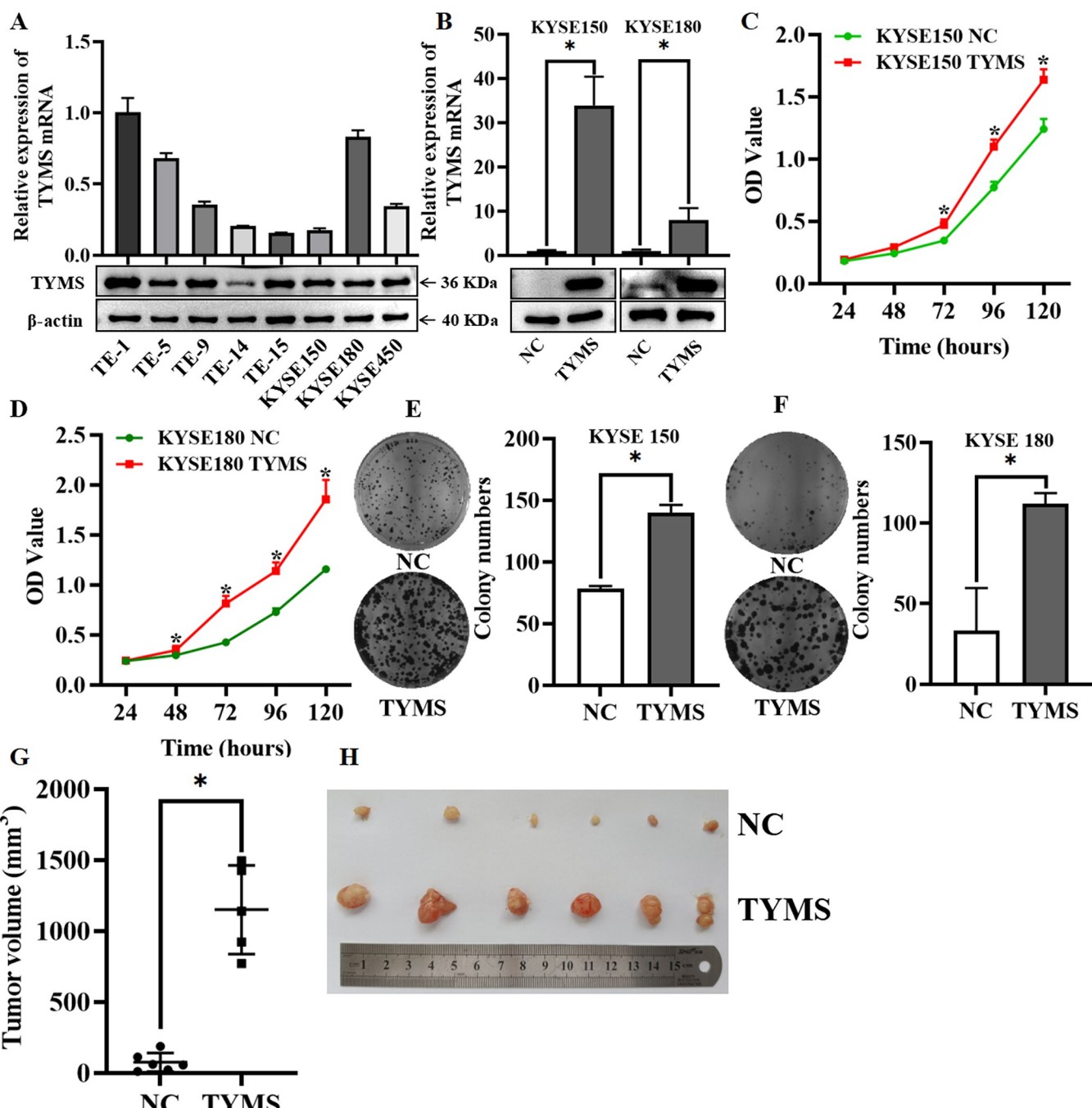

**Fig 1. TYMS promoted ESCC cell proliferation and tumor growth.** (A). RT-qPCR and Western blotting detected the expression of endogenous TYMS in different ESCC cell lines. (B) The overexpression efficiency of TYMS in KYSE150 and KYSE180 cells was detected through qRT-PCR and Western blotting. (C-D) MTT detected the cell proliferation ability after overexpression of TYMS in KYSE150 and KYSE180. (E-F) Clonal formation experiment detected the clonal formation ability after overexpression of TYMS in KYSE150 and KYSE180. (G-H) Subcutaneous tumor formation verified that TYMS promoted tumor growth, and tumor volume was statistically analyzed. Data were representative of three independent experiments and presented as mean ± S.D., and a *p* value < 0.05 was regarded as statistically significant.

treatment groups significantly decreased compared to that of the control groups (Fig 3A and 3B), and the reduced antioxidant capacity (Fig 3C and 3D) re-increased the oxidative stress levels (Fig 3E), resulting in the slowing-down of cancer cell proliferation (Fig 3F and 3G).

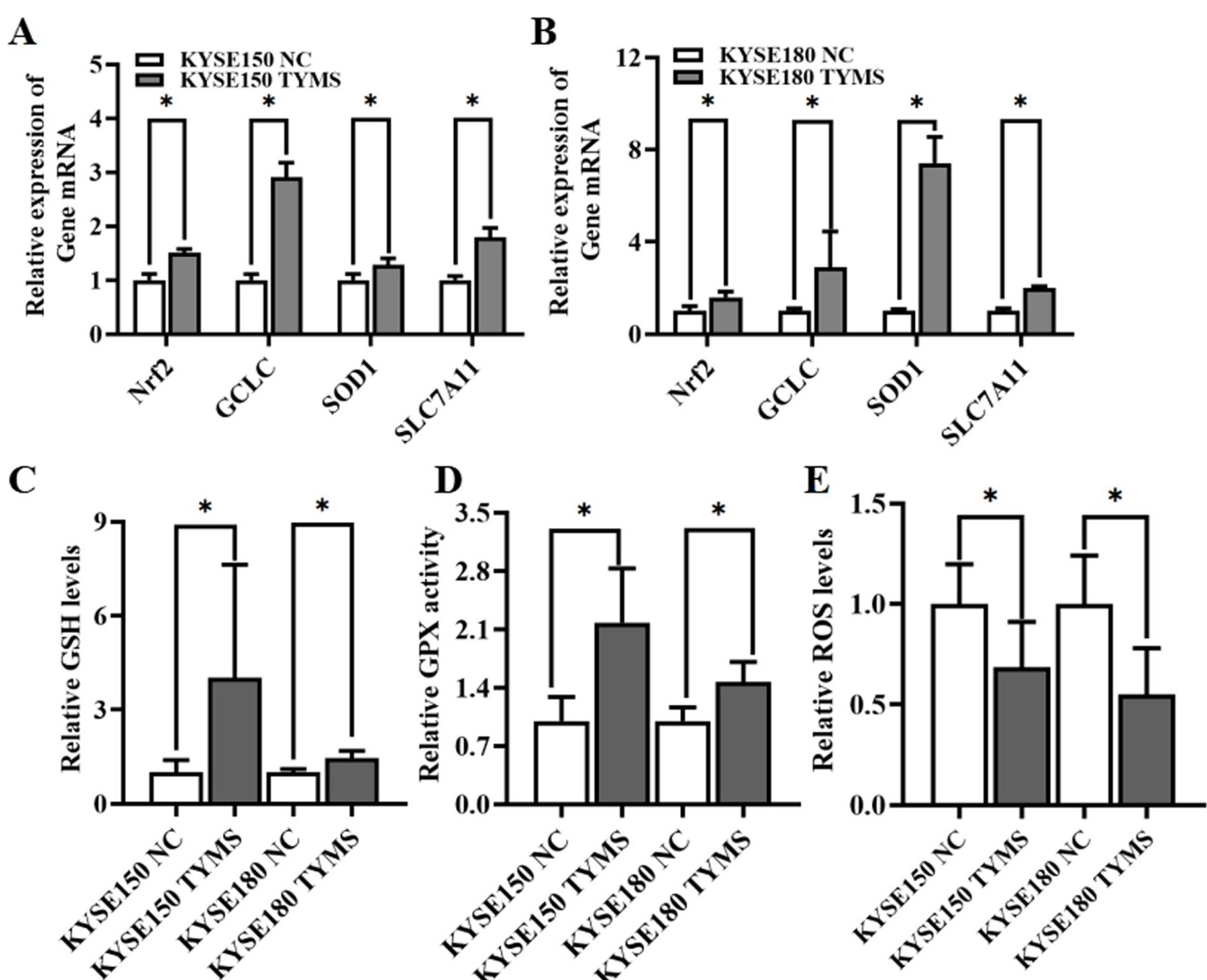

**Fig 2. Increased antioxidant capacity relieved oxidative stress in TYMS-overexpression cells.** (A-B) qRT-PCR detected the expression of Nrf2 and Nrf2 dependent antioxidant enzymes in TYMS-overexpression cells. GSH levels (C), GPX activity (D), and ROS levels (E) in TYMS-overexpression cells and corresponding control cells. Data were representative for three independent experiments and represented as mean ± S.D., a $p$ value < 0.05 was regarded statistically significant.

## Discussion

TYMS, a key enzyme in intracellular nucleic acid metabolism, is involved in the malignant process of multiple cancers. TYMS overexpression is a general event in human carcinogenesis, and is closely related to cancer cell functions including proliferation, invasion, and cell division. Song et al. [18] reported the up-regulation of TYMS in breast cancer tissue, and the increased TYMS promoted cell proliferation of breast cancer and colorectal cancer by accelerating the cell cycle progress [19, 20]. However, Mingxu Fu et al. [21] found promoted proliferation, migration, invasion and reduced cell apoptosis in TYMS-knockdown HeLa cells, and TYMS plays the role of a tumour suppressor gene in cervical cancer. These reveal the differences of TYMS expression and functions in different types of tumors. However, the clinical expression, biological function and carcinogenic mechanism of TYMS in ESCC are still

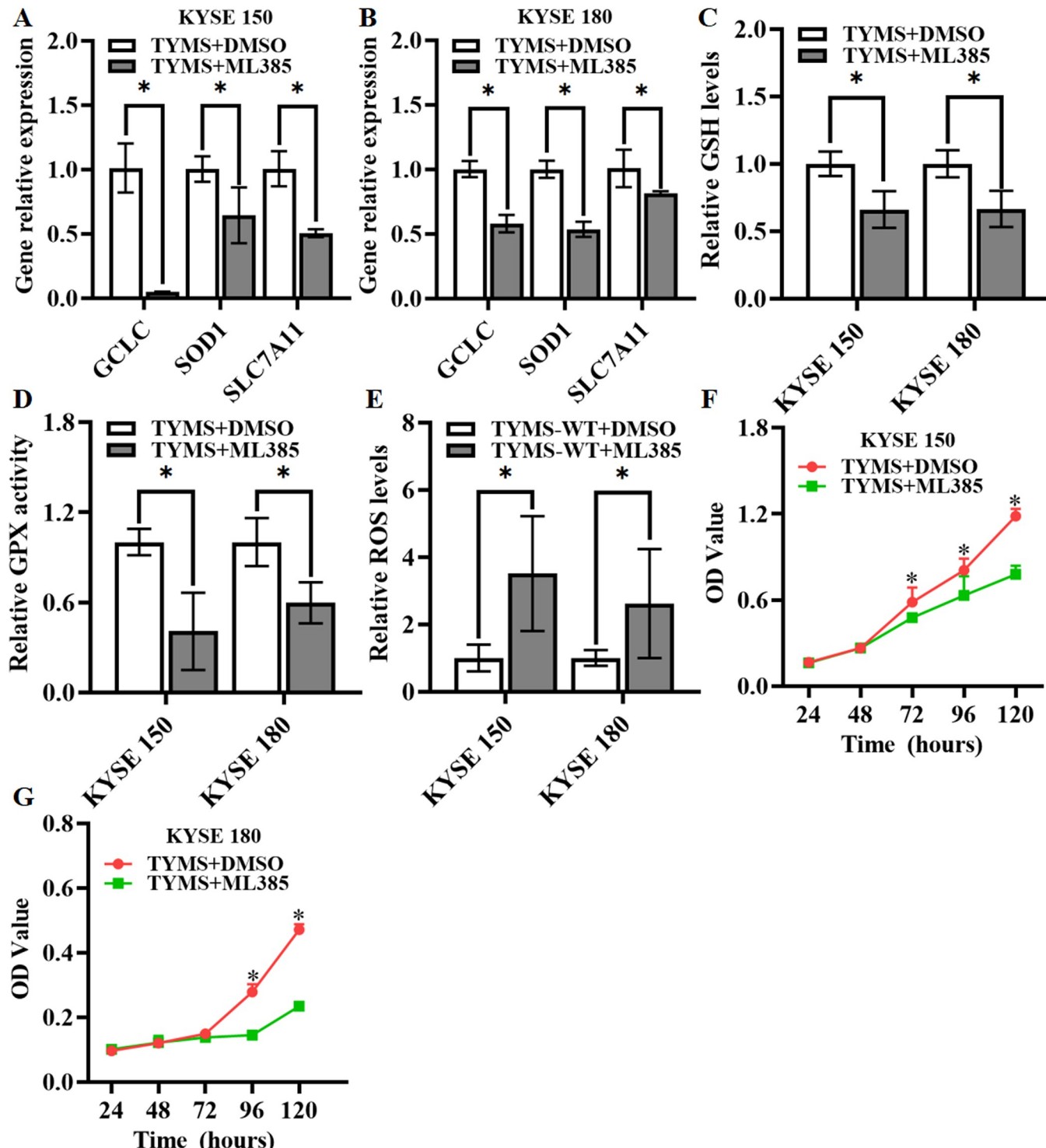

**Fig 3. Analysis of cell viability and oxidative stress after treatment with ML385.** (A-B) Expression analyse of Nrf2 dependent antioxidant enzymes in TYMS-overexpression cells treated with ML385 for 72h. GSH levels (C), GPX activity (D) and ROS levels (E) in TYMS-overexpression cells treated with ML385 for 72h. MTT assay was performed after treatment with ML385 in the TYMS-overexpression cells (F and G). All data are expressed as mean ± standard deviation (SD; three independent experiments), and statistical significance is denoted by * between different groups ($p < 0.05$).

unclear. In the present study, the analysis of GEPIA online database found that the expression of TYMS gene was generally up-regulated in human esophageal cancer (S1 Fig), indicating that detection of TYMS expression could be used as an indicator for ESCC diagnosis. Kimura et al. [22] found that ESCC patients with high TYMS mRNA expression levels had a significantly shorter survival after surgery, and none of the patients received chemotherapy or radiation therapy prior to or following surgery. The non-small cell lung cancer (NSCLC) patients with low TYMS expression also had statistically significantly longer overall survival (OS) and progression-free survival (PFS) compared to those with high TYMS [23], and low TYMS protein expression is a favorable predictive factor for better OS/PFS in NSCLC patients. The large prospective analysis from Niedzwiecki et al. [24] showed that the high tumor TYMS levels were associated with improved OS and disease-free survival (DFS) following an adjuvant therapy for colon cancer. Collectively, these studies demonstrates that the TYMS level in tumor tissues may be a useful marker to predict the postoperative OS of cancer patients, and chemoradiotherapy after surgery could significantly improve the prognosis of cancer patients with high TYMS expression. However, the sample size in these studies was limited. So it is necessary to expand the sample size verification to further confirm the relationship between the expression of TYMS and radiotherapy, chemotherapy, prognosis, and to provide theoretical basis and molecular basis of TYMS for clinical prognostic indicators and drug use.

In order to confirm the function of TYMS in ESCC, TYMS gene was over-expressed in KYSE150 and KYSE180 cells, and its functions were verified by *in vitro* and *in vivo*. The results showed that the overexpression of TYMS promoted the proliferation of ESCC cells and tumor growth, which was consistent with the function of TYMS in other cancers, [13–16]. In order to further explore the mechanism of TYMS playing a role in promoting cancer, transcriptomic sequencing was carried out, and differentially expressed genes related with antioxidant capacity such as Nrf2 were identify. Nrf2 mainly located in the cytoplasm and could be degraded to maintain low levels in the cell cytoplasm by ubiquitination Kelch ECH-associated protein 1 (Keap1) -mediated ubiquitination [25, 26]. Under oxidative stress, Nrf2 dissociates from Keap1 and is transfers to the nucleus to form a dimer with small Maf protein, and combines with the antioxidant response element [27] to activate the transcription of antioxidant genes and defends against oxidative stress [28]. Through the transcriptional regulation expression of glutamate cysteine ligase complex: catalytic subunit (GCLC) and modified subunit (GCLM), Nrf2 could finally realize the regulation of intracellular GSH level [29]. SLC7A11 also is regulated by Nrf2, and is involved in the antioxidant response by transporting cystine into the cells for the production of cysteine and GSH [30]. In the defense against oxidative stress of cells, GSH is one of the major intracellular antioxidants. It could neutralize reactive oxygen species by the synthesis of oxidized glutathione disulfide (GSSG), and also is a key synergistic factor for GPX to catalyze the conversion of peroxides to alcohols. In present study, the up-regulated Nrf2 and Nrf2-dependent enzymes genes were observed in TYMS overexpression cells, and the detection of GSH levels and GPX activities further confirmed the important regulatory role of Nrf2 in glutathione system. Nrf2 participates in regulation of GCLC, SOD1, SCLC7A11 genes expressions and GSH, GPX levels to prevent excessive ROS accumulation and maintain redox homeostasis.

## Conclusion

The present study identifies TYMS as cancer-promoting genes by promoting cell proliferation in human ESCC, and reveals the activating mechanism of the antioxidant protection system by up-regulating Nrf2 and Nrf2-dependent enzymes genes expression to reduce the level of intracellular oxidative stress (Fig 4). The new role of TYMS in intracellular antioxidant

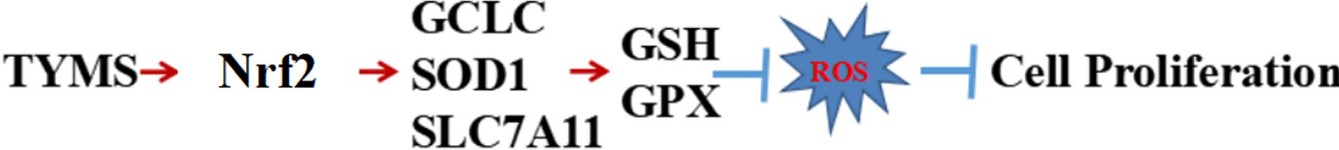

**Fig 4. Regulation model of TYMS on cell proliferation in ESCC, arrows (→) and truncated lines (—|) indicating promoting and inhibiting effects, respectively.**

function suggests that TYMS and Nrf2 could be used as a valuable target to improve the clinical efficacy in ESCC chemotherapy.

## Supporting information

**S1 Fig. TYMS mRNA expression analysis in esophageal cancer tissues (n = 182) and normal tissues (n = 286) from the GEPIA online database.**
(TIF)

**S2 Fig. Silencing TYMS inhabits ESCC cell proliferation.** (A) Knockdown efficiency of TYMS in KYSE180 and TE-1 cells was detected through qRT-PCR and Western blot. MTT (B) and clonal formation experiment (C) detected the cell proliferation ability after knockdown of TYMS in KYSE180 and TE-1. (D and E) qRT-PCR detected the expression of Nrf2 and Nrf2 dependent antioxidant enzyme genes in TYMS-knockdown cells. GSH levels (F), GPX activity (G), and ROS levels (H) in TYMS-knockdown cells and correspondence control cells. Data were representative of least three independent experiments and were presented as mean ± S. D., where a $p$ value $< 0.05$ was regarded as statistically significant.
(TIF)

**S1 Raw images. Full western blots supporting Fig 1A and 1B and S2A Fig.**
(PDF)

## Acknowledgments

We would like to thank Dr. Hans-Uwe Dahms for language editing of a manuscript draft.

## Author Contributions

**Conceptualization:** Jian Yang.

**Data curation:** Jingjing Zhang, Jingtian Chen, Hao Shen.

**Formal analysis:** Hui Sun.

**Funding acquisition:** Jian Yang.

**Investigation:** Jian Yang, Jingjing Zhang, Jingtian Chen, Xiaolong Yang, Hui Sun, Zhenxiang Zhao.

**Methodology:** Jingjing Zhang, Jingtian Chen, Xiaolong Yang, Hui Zhou, Hao Shen.

**Project administration:** Jingjing Zhang.

**Resources:** Jingjing Zhang, Hao Shen.

**Software:** Jingtian Chen, Hui Sun, Zhenxiang Zhao, Hao Shen.

**Supervision:** Jian Yang, Hui Sun, Zhenxiang Zhao.

**Writing – original draft:** Jian Yang, Hui Sun, Hui Zhou.

**Writing – review & editing:** Jian Yang.

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
