## [Decision Letter · Decision Letter 0]

21 Mar 2023

PONE-D-23-05091Thymidylate synthase promotes esophageal squamous cell carcinoma growth by relieving oxidative stress through activating nuclear factor erythroid 2-related factor 2 expressionPLOS ONE

Dear Dr. Yang,

Thank you for submitting your manuscript to PLOS ONE. After careful consideration, we feel that it has merit but does not fully meet PLOS ONE’s publication criteria as it currently stands. Therefore, we invite you to submit a revised version of the manuscript that addresses the points raised during the review process.

Two experts read your manuscript thoroughly and evaluated the merit for publication in PLOS ONE.  As you can find their constructive comments below, they found your work interesting but also noticed several important issues and concerns to be clarified or addressed before further consideration of your work for publication in PLOS ONE.  I see that the reviewers are in general agreement with their major concerns, such as the lack of appropriate gene expression analyses of TYMS and Nrf2 along with TYMS knockdown approaches, the lack of the molecular mechanism through which TYMS upregulates Nrf2, and author's interpretation not fully supported by the data.  In addition, both reviewers have more specific comments and critiques on almost all figures.  I believe that all comments and critiques from the reviewers are critical and important for improvement of your manuscript.  You will need to set up new experiments to address their comments.  Below you can find reviewer's specific comments and suggestions to improve your manuscript. Please ensure that your decision is justified on PLOS ONE’s publication criteria and not, for example, on novelty or perceived impact.

We look forward to receiving your revised manuscript.

Kind regards,

Yoshiaki Tsuji

Academic Editor

PLOS ONE

Journal Requirements:

4. We note that you have stated that you will provide repository information for your data at acceptance. Should your manuscript be accepted for publication, we will hold it until you provide the relevant accession numbers or DOIs necessary to access your data. If you wish to make changes to your Data Availability statement, please describe these changes in your cover letter and we will update your Data Availability statement to reflect the information you provide

Reviewers' comments:

Reviewer's Responses to Questions

**Comments to the Author**

1. Is the manuscript technically sound, and do the data support the conclusions?

Reviewer #1: Partly

Reviewer #2: Yes

2. Has the statistical analysis been performed appropriately and rigorously? 

Reviewer #1: I Don't Know

Reviewer #2: No

3. Have the authors made all data underlying the findings in their manuscript fully available?

Reviewer #1: No

Reviewer #2: Yes

4. Is the manuscript presented in an intelligible fashion and written in standard English?

Reviewer #1: Yes

Reviewer #2: Yes

5. Review Comments to the Author

Reviewer #1: In this study, the Authors proposed an interesting role of TYMS in regulating esophageal squamous cell carcinoma through the activation of NRF-2. The proposed studies partially support this proposal, but some methodological and technical issues limit the reliability of the conclusions. Some issues need clarification before this manuscript can be recommended for publication.

General comments:

1). My main doubts relate to the analysis of TYMS in the context of cell proliferation. First, since the authors decided to analyze the panel of ESCC lines, why was only the mRNA level shown and not the protein level? As it is well known, both indicators do not have to correlate, and from the point of view of the thesis that the authors want to prove, the actual level of TYMS in cells is more important. In addition, in order to prove their thesis, the authors should analyze the proliferation rate of more cell lines and analyze whether the increase in proliferation correlates with the level of TYMS protein in these cells.

2). Secondly, the authors' results showing different prognoses for patients with high TYMS levels depending on the treatments performed after surgery are very interesting but may confuse readers. Therefore, the authors should try to explain this phenomenon, especially in the context of other works, such as 10.3892/ol.2010.227, and other types of cancer in which a high level of TYMS is observed.

3). The discussion of the results should be much more detailed. As a rule, each result should be discussed, and each Figure should be cited in the text (no citation for Fig. 1C, D, and E; only general citation for Figs. 3 and 4).

4). In addition, since the authors are analyzing the role of TYMS in cancer progression, other recent papers related to this topic should also be discussed (colon cancer, breast and lung cancer). Moreover, data reported by Kimura et al. about TYMS dehydrogenase mRNA levels in esophageal cancer should be interpreted.

Specific comments:

5). Provide detailed information about statistical analysis in Figure legends.

6). In lines 243-244, the Authors stated that "[…] cell lines with lower TYMS mRNA were chosen for TYMS over-expression experiment". However, KYSE180 cells revealed a relatively high TYMS mRNA level. Why did the authors choose this cell line, not those with much lower TYMS expression, such as TE-9 - TE-15?

7). In order to confirm the hypothesis of the role of TYMS in the regulation of proliferation via the NRF-2-ROS pathway, it is necessary to silence TYMS in cells with high protein levels (for example, HET-1A or TE-1) and show that the proliferation and oxidative potential of the cell decreases (level genes, ROS level, etc.).

8). More detailed information on the methodology should be given. E.g., I couldn't find any detailed information about the procedure for stripping and re-probing a membrane. What was the average quality of the RNA extracted (A230/280 ratio)? Why was only one housekeeping gene selected? Has any analysis been performed based on any software (for example geNorm or others)? Please provide the sequence of the primers used (The Table 1 is missing).

9). Calling clones of TYMS overexpressing cell lines WT may be misleading as this designates Wild Type. In order to avoid confusing the readers, I suggest changing it, e.g., KYSE-Tyms.

10).ML385 treatment. First, it is an NRF-2 inhibitor, not a TYMS, as the authors presented. Secondly, what was its concentration, because rather not 20 mg/mL as indicated in the manuscript (10 mM corresponds to 5.11 mg/mL). Why did the authors decide to incubate the cells with the inhibitor for such a long time (96h)?

11). It is known that the MTT test is based on reducing a tetrazolium salt to formazan crystals. It is a process that is sensitive to oxidative changes that may affect the results obtained. If so, why did the Authors choose MTT when studying the TYMS-NRF-2-ROS pathway and not a more indifferent test like neutral red uptake or SRB?

12). Figure 4 – Figure A is ESCA instead of ESCC. Figure B please provide a better quality of images. Figure D and E– there must be a mistake; survival time ends at 1200 months (100 years).

13). Additionally, please provide the statistical analysis for proliferation curves.

Reviewer #2: Yang et al. investigated the clinical values of thymidylate synthase (TYMS) and its potential role in regulating carcinogenesis of esophageal squamous cell cancer (ESCC). The study is interesting in a way that the authors found that overexpression of TYMS in ESCC cells may activate gene expression of NRF2 and NRF2-dependent antioxidant enzymes, and this may in turn relieve ROS-dependent oxidative stress and promote ESCC cell growth. However, the experimental design and setup are not robust and comprehensive, and most of the critical results are not convincing. These unfortunately weaken the significance of this study.

1. In the “Abstract” and the first subheading "TYMS expression and prognosis analysis of esophageal cancer ", the authors mentioned “TMSB10”, which encodes thymosin beta 10, in their study. However, the reviewer was not clear and confusing what is its relevance to this study.

2. Figure 1— Although the authors indicated that the survival analysis in 58 ESCC patients with radiotherapy and chemotherapy after surgery showed opposite results to the survival analysis in 97 ESCC patients without radiotherapy and chemotherapy after surgery (Fig. 1D, 1E), there was not any explanation for this discrepancy. Moreover, there was no explanation for the correlation between high TYMS expression and a better patient prognosis shown in Fig. 1B, which actually conflicts to the role of TYMS in ESCC carcinogenesis. The acronyms “ESCA” and “ESCC” should be spelled out in the figure legend.

3. Figure 2— As TYMS may be highly expressed in ESCC cell lines, it would be better to select a high TYMS-expressing ESCC cell line and knockdown of TYMS in this cell line for a comparative investigation. It is not clear why the authors chose KYSE180 for TYMS-overexpression experiments as this ESCC cell line express relatively higher levels of TYMS compared to the other ESCC cell lines (Fig. 2A, 2C).

4. Figure 3— It is still unclear why overexpression of TYMS causes upregulation of NFE2L2 in ESCC cells. The reviewer wonders if knockdown of TYMS could downregulate NFE2L2 expression in ESCC cell lines. Furthermore, the NFE2L2-knockdown experiments were missing for confirming the effects of NFE2L2 upregulation in TYMS-induced ESCC cell proliferation.

5. Figures 2D, 2E, 4A, 4B — there was no statistical comparison between control and experiment groups.

6. It would be more interesting and impactful to study the mechanisms underlying the depletion of TYMS mRNA in high TYMS-expressing ESCC cell lines, such as to examine the gene expression of NRF2 and NRF2-dependent antioxidant enzymes and their functional regulation in ROS homeostasis and ESCC cell growth in cell-based and animal models.

6. PLOS authors have the option to publish the peer review history of their article (what does this mean?). If published, this will include your full peer review and any attached files.

Reviewer #1: No

Reviewer #2: No

---

## [Author Response · Author response to Decision Letter 0]

17 May 2023

Dear Editor

We want to thank you for your useful comments and suggestions on our manuscript. We have revised the manuscript accordingly, and detailed corrections are listed below:

Comments of Editor:

Two experts read your manuscript thoroughly and evaluated the merit for publication in PLOS ONE. As you can find their constructive comments below, they found your work interesting but also noticed several important issues and concerns to be clarified or addressed before further consideration of your work for publication in PLOS ONE. I see that the reviewers are in general agreement with their major concerns, such as the lack of appropriate gene expression analyses of TYMS and Nrf2 along with TYMS knockdown approaches, the lack of the molecular mechanism through which TYMS upregulates Nrf2, and author's interpretation not fully supported by the data. In addition, both reviewers have more specific comments and critiques on almost all figures. I believe that all comments and critiques from the reviewers are critical and important for improvement of your manuscript. You will need to set up new experiments to address their comments. Below you can find reviewer's specific comments and suggestions to improve your manuscript.

Author Response:

Thanks a lot for your suggestion and comments. Your suggestion is very necessary for the present study to clear the mechanism of overexpressed TYMS on Nrf2. Before submission, we tried to establish a TYMS stable knockout cell lines in TE-1 or other cell lines using Lentivirus Vector. However, it was hard to construct TYMS-silencing cells, and we fail until the paper submitted to the journal. Fortunately, we finally succeed to construct TYMS-silencing cells during the review period, and also added relevant experiments (Cell proliferation assays, Gene expression analysis, Detection of GSH, ROS, and GPx) in TYMS-silencing cells. The related results could be found in S1 Figure. The Nrf2-knockdown experiments were also difficult. ML385, an NRF-2 inhibitor, was used to replace the Nrf2-knockdown experiments, and confirmed that ML385 treatment in TYMS-overexpression cells re-increased the oxidative stress levels and the slowing-down cancer cell proliferation. Moreover, the comments and critiques from the reviewers are critical and important on the present manuscript, and we are very grateful for their generous help. According to the reviewer's suggestions, we carefully revised the manuscript, and also response to their questions point by point. Thanks again.

Journal Requirements:

1). Please ensure that your manuscript meets PLOS ONE's style requirements, including those for file naming. The PLOS ONE style templates can be found at https://journals.plos.org/plosone/s/file?id=wjVg/PLOSOne_formatting_sample_main_body.pdf and https://journals.plos.org/plosone/s/file?id=ba62/PLOSOne_formatting_sample_title_authors_affiliations.pdf

Author Response: 

Thank you for your suggestions, and modifications including format among others of Abstract, Introduction, Materials and Methods have been made according to PLOS ONE's style requirements. 

2). PLOS ONE now requires that authors provide the original uncropped and unadjusted images underlying all blot or gel results reported in a submission’s figures or Supporting Information files. This policy and the journal’s other requirements for blot/gel reporting and figure preparation are described in detail at https://journals.plos.org/plosone/s/figures#loc-blot-and-gel-reporting-requirements and https://journals.plos.org/plosone/s/figures#loc-preparing-figures-from-image-files. When you submit your revised manuscript, please ensure that your figures adhere fully to these guidelines and provide the original underlying images for all blot or gel data reported in your submission. See the following link for instructions on providing the original image data: https://journals.plos.org/plosone/s/figures#loc-original-images-for-blots-and-gels. 

Author Response: 

Thank you for your suggestions. According to your requirements, we provided the original uncropped and unadjusted images underlying all blots which could be found in Supporting Information files and named as S2 raw images (S2 Fig. Full western blots supporting Fig 2A-B and S1 Fig A). We also noted that these blots in Supporting Information in our cover letter.

3).We note that you have indicated that data from this study are available upon request. PLOS only allows data to be available upon request if there are legal or ethical restrictions on sharing data publicly. 

We note that you have stated that you will provide repository information for your data at acceptance. Should your manuscript be accepted for publication, we will hold it until you provide the relevant accession numbers or DOIs necessary to access your data. If you wish to make changes to your Data Availability statement, please describe these changes in your cover letter and we will update your Data Availability statement to reflect the information you provide

Author Response: 

Thank you for your comments. After careful consideration, we decided to make changes to our Data Availability statement. All data are fully available without restrictions, and All relevant data are within the manuscript and its Supporting Information files. Thanks again.

Comments of Reviewer #1

1). My main doubts relate to the analysis of TYMS in the context of cell proliferation. First, since the authors decided to analyze the panel of ESCC lines, why was only the mRNA level shown and not the protein level? As it is well known, both indicators do not have to correlate, and from the point of view of the thesis that the authors want to prove, the actual level of TYMS in cells is more important. In addition, in order to prove their thesis, the authors should analyze the proliferation rate of more cell lines and analyze whether the increase in proliferation correlates with the level of TYMS protein in these cells.

Author Response:

Thanks a lot for your suggestion and comments. Indeed, there is no necessary connection between mRNA and protein, and it is necessary to detect the levels of TYMS protein in Multiple esophageal squamous cell carcinoma cell lines. TYMS protein expression in ESCC cell lines was detected through Western blot (WB), and Related results could be found in Figure 2A. According to the results of WB and Q-PCR, both te-5 and te-14 with low TYMS were firstly used for the overexpression of TYMS. However, we failed to construct the stable transgenic cell line with TYMS overexpression in te-5 and te-14, and other cell lines had to used for the constructing. Finally, constructing of TYMS overexpression was successed in TYMS150 and 180, and both cell lines were used for the future experiments (such as MTT, clone formation assay, and so on). Thanks again.

2). Secondly, the authors' results showing different prognoses for patients with high TYMS levels depending on the treatments performed after surgery are very interesting but may confuse readers. Therefore, the authors should try to explain this phenomenon, especially in the context of other works, such as 10.3892/ol.2010.227, and other types of cancer in which a high level of TYMS is observed.

Author Response:

Thanks a lot for your comments. The study (10.3892/ol.2010.227) provided from you is necessary to prove our view. The patients with low TYMS expression exhibited a worse prognosis from GEPIA online database (Fig 1B), and the same results could be found in survival analysis of ESCC patients with radiotherapy and chemotherapy (Fig 1E). However, Kimura et al. [22] found that ESCC patients with high TYMS mRNA expression levels had a significantly shorter survival after surgery, and none of the patients received chemotherapy or radiation therapy prior to or following surgery. In present study, the survival analysis in ESCC patients without radiotherapy and chemotherapy after surgery showed similar results (Fig 1D). The large prospective analysis from Niedzwiecki et al.[23] showed that the high tumor TYMS levels were associated with improved overall survival (OS) and disease-free survival (DFS) following adjuvant therapy for colon cancer. The non-small cell lung cancer (NSCLC) patients with low TYMS expression also had statistically significantly longer OS and progression-free survival (PFS) than those with high TYMS, and low TYMS protein expression is a favorable predictive factor for better OS/PFS in NSCLC patients [24]. Collectively, this study demonstrates that the TYMS level in tumor tissues may be a useful marker to predict the postoperative OS of cancer patients, and the use of anti-tumor drugs (such as 5-fluorouracil) could significantly improve the prognosis of ESCC patients with high TYMS expression. These could be found in revised manuscript in page 11. Thanks again for your help.

3). The discussion of the results should be much more detailed. As a rule, each result should be discussed, and each Figure should be cited in the text (no citation for Fig. 1C, D, and E; only general citation for Figs. 3 and 4).

Author Response:

Thanks a lot for your comments. Your suggestions are absolutely right. According to your suggestion, we checked the paper, and also cited all figure in correct position. 

Discussions related to the results were also supplemented, which could be found in the Results and Discussion of revised manuscript. Thanks again for your suggestions.

4). In addition, since the authors are analyzing the role of TYMS in cancer progression, other recent papers related to this topic should also be discussed (colon cancer, breast and lung cancer). Moreover, data reported by Kimura et al. about TYMS dehydrogenase mRNA levels in esophageal cancer should be interpreted.

Author Response:

Thanks very much for your comments. The role of TYMS in cancer progression was discussed, and some related study also were cited. The detailed content is as follows: The patients with low TYMS expression exhibited a worse prognosis from GEPIA online database (Fig 1B), and the same results could be found in survival analysis of ESCC patients with radiotherapy and chemotherapy (Fig 1E). However, Kimura et al. [22] found that ESCC patients with high TYMS mRNA expression levels had a significantly shorter survival after surgery, and none of the patients received chemotherapy or radiation therapy prior to or following surgery. In present study, the survival analysis in ESCC patients without radiotherapy and chemotherapy after surgery showed similar results (Fig 1D). The large prospective analysis from Niedzwiecki et al.[23] showed that the high tumor TYMS levels were associated with improved overall survival (OS) and disease-free survival (DFS) following adjuvant therapy for colon cancer. The non-small cell lung cancer (NSCLC) patients with low TYMS expression also had statistically significantly longer OS and progression-free survival (PFS) than those with high TYMS, and low TYMS protein expression is a favorable predictive factor for better OS/PFS in NSCLC patients [24]. Collectively, this study demonstrates that the TYMS level in tumor tissues may be a useful marker to predict the postoperative OS of cancer patients, and the use of anti-tumor drugs (such as 5-fluorouracil) could significantly improve the prognosis of ESCC patients with high TYMS expression. Thanks again for your suggestion.

5). Provide detailed information about statistical analysis in Figure legends.

Author Response:

Thanks very much for your comments. According to your suggestions, information about statistical analysis was modified and as follows: Data were presented with mean ± standard deviation (S.D.) from least three independent experiments. All statistical analysis was performed using SPSS software (version 17.0, SPSS Inc., Chicago, IL, USA), and data graphs were generated using GraphPad Prism 5 Software (GraphPad Software Inc. La Jolla, CA, USA). Survival curves were constructed using the Kapla-Meier method and differences in survival were evaluated using the log-rank test. Data homogeneity of variance and normality were tested using the Levene tests and Kolmogorov–Smirnov, respectively. p values were performed using Student’s t-test and one-way ANOVA analysis between groups, and a value of p < 0.05 was considered as statistically significant which was denoted with *. Besides, more detailed information about statistical analysis was provided in Figure legends, which could be found in revised manuscript. Thanks again.

6). In lines 243-244, the Authors stated that "[…] cell lines with lower TYMS mRNA were chosen for TYMS over-expression experiment". However, KYSE180 cells revealed a relatively high TYMS mRNA level. Why did the authors choose this cell line, not those with much lower TYMS expression, such as TE-9 - TE-15?

Author Response:

Thanks very much for your comments. In present study, TYMS protein expression in ESCC cell lines was detected through qRT-PCR and Western blot (WB), and Related results could be found in Figure 2 A. According to the results of WB and qRT-PCR, both te-5 and te-14 with low TYMS were firstly used for the overexpression of TYMS. However, we failed to construct the stable transgenic cell line with TYMS overexpression in te-5 and te-14, and other cell lines had to used for the constructing. Finally, the stable TYMS expression cell line was constructed successfully in TYMS150 and 180, and the efficiency in TYMS over-expression cells were also confirmed by qRT-PCR and Western blot (Fig 2B). Thanks again.

7). In order to confirm the hypothesis of the role of TYMS in the regulation of proliferation via the NRF-2-ROS pathway, it is necessary to silence TYMS in cells with high protein levels (for example, HET-1A or TE-1) and show that the proliferation and oxidative potential of the cell decreases (level genes, ROS level, etc.).

Author Response:

Thanks very much for your comments. Your viewpoint is correct, and it is necessary to silence TYMS. In the early stages of this study, we tried to establish a TYMS stable knockout cell lines in TE-1 and TYMS180 using Lentivirus Vector. It is hard to construct TYMS-silencing cells, and we fail until the paper submitted to the journal. Fortunately, we finally succeed to construct TYMS-silencing cells, and the related rsults could be found in Figure S1. Thanks again.

8). More detailed information on the methodology should be given. E.g., I couldn't find any detailed information about the procedure for stripping and re-probing a membrane. What was the average quality of the RNA extracted (A230/280 ratio)? Why was only one housekeeping gene selected? Has any analysis been performed based on any software (for example geNorm or others)? Please provide the sequence of the primers used (The Table 1 is missing).

Author Response:

Thanks very much for your comments. According to your suggestion, more detailed information about qRT-PCR analysis was provided, and as follows: Total RNA was extracted from 5.0 x 105 cells using the RNA extraction kit according to the manufacturer's instructions. The concentration of isolated RNA extracted were quantified at 230nm, 260nm, and 280nm using NanoDrop Spectrophotomete (Thermo Scientific, USA), and only those samples with a 260nm /280 nm ratio between 1.8-2.1 and a 260nm /230 nm ratio of more than 2.0 were used for further analysis. Using MMLV reverse transcriptase (Takara Shuzo Co. Ltd., Kyoto, Japan), the first-strand cDNA was synthesized from the 200 ng of isolated total RNA. Then, the expression of gene mRNA was determined using the SYBR® Green PCR Master Mix (Vazyme Biotech Co., Ltd) by ABI 7500 Real-Time PCR System (Applied Biosystems, Foster City, CA, USA). The qRT-PCR specific primers were designed based on genes sequences from the NCBI GenBank database using Primer software (Premier Biosoft, Palo Alto, CA, USA), and were shown in Table 1. The qRT-PCR reaction mixture was shown below: 0.5 μL each forward and reverse primers (10 μM), 1 μL cDNA template, 10 μL Realtime PCR Super mix (2×), and RNase free dH2O to adjust to 20 μL. Using the 2-ΔΔCt method, the relative quantification of genes mRNA expression was achieved by concurrent amplification of the GAPDH endogenous control. These could be found in page 7.

9). Calling clones of TYMS overexpressing cell lines WT may be misleading as this designates Wild Type. In order to avoid confusing the readers, I suggest changing it, e.g., KYSE-Tyms.

Author Response:

Thanks very much for your comments. Indeed, the word “WT ” might be misleading, and all “KYSE-WT” in figures or paper was replaced with TYMS in revised manuscript.

10).ML385 treatment. First, it is an NRF-2 inhibitor, not a TYMS, as the authors presented. Secondly, what was its concentration, because rather not 20 mg/mL as indicated in the manuscript (10 mM corresponds to 5.11 mg/mL). Why did the authors decide to incubate the cells with the inhibitor for such a long time (96h)?

Author Response:

Thanks very much for your comments. We are so sorry for the mistakes due to our careless. After confirmation, the stock solutions of ML385 were 5mM, and cells were treated with 5.0µM ML385. After that the cells were added with ML385, the significant difference was not observed in 24h or 36h. After a series of pre tests, 72 h rather than 96h was chosen as the action time for the further experiments. We apologize again for our carelessness．Thanks again for your help.

11). It is known that the MTT test is based on reducing a tetrazolium salt to formazan crystals. It is a process that is sensitive to oxidative changes that may affect the results obtained. If so, why did the Authors choose MTT when studying the TYMS-NRF-2-ROS pathway and not a more indifferent test like neutral red uptake or SRB?

Author Response:

Thanks very much for your comments. Your suggestion is very professional. As you know, MTT assay is a routine method for detecting cell proliferation, and we overlooked this point when designing experimental plans. There might be some effect of changed ROS on MTT test, and we try to minimize these impacts as much as possible (such as control group and experimental group was added with an equal amount of excessive MTT). Your suggestion is very correct, and We will take your suggestion into consideration in further experiment. Thanks again.

12). Figure 4- Figure A is ESCA instead of ESCC. Figure B please provide a better quality of images. Figure D and E there must be a mistake; survival time ends at 1200 months (100 years).

Author Response:

Thanks a lot for your comments. After a careful check, we are very sorry that the ESCA was mistakenly written as ESCC, which had been corrected. Figure B was replaced with a better quality of images. The mistake in Figure D and E has been revised, and the day (survival time) was mistakenly written with months.

13). Additionally, please provide the statistical analysis for proliferation curves.

Author Response:

Thanks a lot for your comments. The statistical analysis for proliferation curves was provided in Statistics analysis of revised manuscript, and as follows: p values were performed using Student’s t-test between groups, and a value of p < 0.05 was considered as statistically significant which was denoted with *. The significant difference had bee marked with * in all proliferation curves.

Comments of Reviewer #2

1). In the “Abstract” and the first subheading "TYMS expression and prognosis analysis of esophageal cancer ", the authors mentioned “TMSB10”, which encodes thymosin beta 10, in their study. However, the reviewer was not clear and confusing what is its relevance to this study.

Author Response:

Thanks a lot for your comments. We are very sorry for the mistakes. Due to our negligence, the word “TYMS” was mistakenly written as “TMSB10”. After careful check, the mistakes had been corrected in revised manuscript. 

2). Figure 1— Although the authors indicated that the survival analysis in 58 ESCC patients with radiotherapy and chemotherapy after surgery showed opposite results to the survival analysis in 97 ESCC patients without radiotherapy and chemotherapy after surgery (Fig. 1D, 1E), there was not any explanation for this discrepancy. Moreover, there was no explanation for the correlation between high TYMS expression and a better patient prognosis shown in Fig. 1B, which actually conflicts to the role of TYMS in ESCC carcinogenesis. The acronyms “ESCA” and “ESCC” should be spelled out in the figure legend.

Author Response:

Thanks a lot for your comments. Firstly, the acronyms“ESCA” and “ESCC” was spelled out in the figure legend. Secondly, there was indeed not any explanation for above discrepancy, and more detail information could be found in modified manuscript, and was as follows: The patients with low TYMS expression exhibited a worse prognosis from GEPIA online database (Fig 1B), and the same results could be found in survival analysis of ESCC patients with radiotherapy and chemotherapy (Fig 1E). However, Kimura et al. [22] found that ESCC patients with high TYMS mRNA expression levels had a significantly shorter survival after surgery, and none of the patients received chemotherapy or radiation therapy prior to or following surgery. In present study, the survival analysis in ESCC patients without radiotherapy and chemotherapy after surgery showed similar results (Fig 1D). The large prospective analysis from Niedzwiecki et al.[23] showed that the high tumor TYMS levels were associated with improved overall survival (OS) and disease-free survival (DFS) following adjuvant therapy for colon cancer. The non-small cell lung cancer (NSCLC) patients with low TYMS expression also had statistically significantly longer OS and progression-free survival (PFS) than those with high TYMS, and low TYMS protein expression is a favorable predictive factor for better OS/PFS in NSCLC patients [24]. Collectively, this study demonstrates that the TYMS level in tumor tissues may be a useful marker to predict the postoperative OS of cancer patients, and the use of anti-tumor drugs (such as 5-fluorouracil) could significantly improve the prognosis of ESCC patients with high TYMS expression.

3). Figure 2— As TYMS may be highly expressed in ESCC cell lines, it would be better to select a high TYMS-expressing ESCC cell line and knockdown of TYMS in this cell line for a comparative investigation. It is not clear why the authors chose KYSE180 for TYMS-overexpression experiments as this ESCC cell line express relatively higher levels of TYMS compared to the other ESCC cell lines (Fig. 2A, 2C).

Author Response: 

Thanks a lot for your comments. In the original paper, the results of mRNA expression by QRT-PCR was showed in Multiple esophageal squamous cell carcinoma cell lines, and TYMS protein expression in ESCC cell lines was further detected through Western blot (WB). The Related results could be found in Figure 2 A. According to the results of WB and Q-PCR, both te-5 and te-14 with low TYMS were firstly used for the overexpression of TYMS. However, we failed to construct the stable transgenic cell line with TYMS overexpression in te-5 and te-14, and other cell lines had to used for the constructing. Finally, constructing of TYMS overexpression was successed in TYMS150 and 180, and both cell lines were used for the future experiments (such as MTT, clone formation assay, and so on). Moreover, it is necessary to silence TYMS as you As you pointed out. In the early stages of this study, we tried to establish a TYMS stable knockout cell lines in TE-1 and TYMS180 using Lentivirus Vector. It is hard to construct TYMS-silencing cells, and we fail until the paper submitted to the journal. Fortunately, we finally succeed to construct TYMS-silencing cells, and the related rsults could be found in Figure S1. Thanks again.

4)Figure 3- It is still unclear why overexpression of TYMS causes upregulation of NFE2L2 in ESCC cells. The reviewer wonders if knockdown of TYMS could downregulate NFE2L2 expression in ESCC cell lines. Furthermore, the NFE2L2-knockdown experiments were missing for confirming the effects of NFE2L2 upregulation in TYMS-induced ESCC cell proliferation.

Author Response: 

Thanks a lot for your comments. Your suggestion is very necessary for the present study to clear the mechanism of overexpressed TYMS on Nrf2. As we mentioned in earlier response, we tried to establish a TYMS stable knockout cell lines in TE-1 and TYMS180 using Lentivirus Vector in the early stages of this study. However, it was hard to construct TYMS-silencing cells, and we fail until the paper submitted to the journal. Fortunately, we finally succeed to construct TYMS-silencing cells, and also added relevant experiments in TYMS-silencing cells. The related rsults could be found in Figure S1. The Nrf2-knockdown experiments were also difficult. ML385, an NRF-2 inhibitor, was used to replace the Nrf2-knockdown experiments, and confirmed that ML385 treatment in TYMS-overexpression cells re-increased the oxidative stress levels and the slowing-down cancer cell proliferation. Thanks again.

5). Figures 2D, 2E, 4A, 4B — there was no statistical comparison between control and experiment groups.

Author Response: 

Thanks a lot for your comments. The statistical comparison between control and experiment groups in Figure 2D, 2E, 4A, 4B was carried out, and the a value of p < 0.05 was considered as statistically significant which was denoted with *.

6). It would be more interesting and impactful to study the mechanisms underlying the depletion of TYMS mRNA in high TYMS-expressing ESCC cell lines, such as to examine the gene expression of NRF2 and NRF2-dependent antioxidant enzymes and their functional regulation in ROS homeostasis and ESCC cell growth in cell-based and animal models.

Author Response: 

Thanks a lot for your comments. Your suggestion is very professional and meaningful. The successful construction of TYMS-silencing cells ensured the smooth progress of the experiment which partly confirmed the hypothesis of the role of TYMS in the regulation of proliferation via the Nrf2-ROS pathway. Thanks again.

Following the comments of reviewers, the manuscript has been revised. We hope that the revised manuscript and the accompanying responses will be sufficient to make it suitable for publication in journal of Plos One. Your kind consideration will be highly appreciated. 

With best wishes. 

Jian Yang

---

## [Decision Letter · Decision Letter 1]

14 Jun 2023

PONE-D-23-05091R1Thymidylate synthase promotes esophageal squamous cell carcinoma growth by relieving oxidative stress through activating nuclear factor erythroid 2-related factor 2 expressionPLOS ONE

Dear Dr. Yang,

Thank you for submitting your manuscript to PLOS ONE. I am still waiting for Reviewer #1 comments but let me share the Reviewer #2 comments to save the time. We encourage and invite you to submit a revised version of the manuscript that addresses the Reviewer #2 points. I may or may not be able to share the Reviewer #1 comments, depending on the timeliness during the review process.

We look forward to receiving your revised manuscript.

Kind regards,

Yoshiaki Tsuji

Academic Editor

PLOS ONE

Journal Requirements:

Reviewers' comments:

Reviewer's Responses to Questions

**Comments to the Author**

1. If the authors have adequately addressed your comments raised in a previous round of review and you feel that this manuscript is now acceptable for publication, you may indicate that here to bypass the “Comments to the Author” section, enter your conflict of interest statement in the “Confidential to Editor” section, and submit your "Accept" recommendation.

Reviewer #2: All comments have been addressed

2. Is the manuscript technically sound, and do the data support the conclusions?

Reviewer #2: Yes

3. Has the statistical analysis been performed appropriately and rigorously? 

Reviewer #2: Yes

4. Have the authors made all data underlying the findings in their manuscript fully available?

Reviewer #2: Yes

5. Is the manuscript presented in an intelligible fashion and written in standard English?

Reviewer #2: No

6. Review Comments to the Author

Reviewer #2: The authors have extensively revised the manuscript based on the reviewers' suggestions except some points (mentioned below) remaining to be further addressed.

For the authors’ response to comments #2:

This part is still a big confusion to the reviewer and maybe to the readers as well. Although the authors found out other reports to support their findings “TYMS promotes ESCC cell growth by relieving oxidative stress through activating Nrf2 expression” (e.g., Kimura et al. ESCC patients’ outcome [ref. 22], NSCLC report [ref.24]), the discrepancy in the relationship between the expression levels of TYMS and the outcome of cancer patients receiving chemotherapy/radiotherapy or not remains to be not addressed or discussed (Figure 1). Actually, although TYMS is highly expressed in ESCC tissue samples in the GEPIA data, the patients with high TYMS expression exhibit a better prognosis (Figures 1A and 1B). The similar trend can be found in Figure 1E, while these patients have received radiotherapy and chemotherapy. In contrast, the survival curve in patients without radiotherapy and chemotherapy indicates that the patients with high TYMS expression exhibit a worse prognosis (Figure 1D), which favors to the authors’ present study and experimental setting, that is “TYMS overexpression promotes ESCC cell growth in the condition without chemotherapy and/or radiotherapy”.

Specific comments and suggestions:

1. The authors have to clearly distinguish the clinical sample data with or without chemotherapy/radiotherapy because these two conditions seem to lead to opposite patients’ outcome. The current description for Figure 1 and the second paragraph in the discussion section kind of mix up these two conditions (i.e. with or without chemotherapy/radiotherapy) which is confusing and does not provide any more new insights.

2. Since chemotherapy and radiotherapy cause an increase in intracellular ROS that can lead to apoptosis via extrinsic or intrinsic pathways and based on the authors’ present study that TYMS activates Nrf2 expression and the following ROS scavenger events, it would be better to discuss the possibility why high TYMS expressing patients receiving chemotherapy and radiotherapy may have a better clinical outcome. Instead, the current discussion for the reports of the clinical outcome of patients who receive chemotherapy and radiotherapy or not would be better to move to the description for Figure 1 and mention that it seems there are two clinical conditions for the roles of TYMS in patients' survival and the author will focus on studying the condition without chemotherapy and radiotherapy.

3. Are the patients in the GEPIA data receiving chemotherapy/radiotherapy or not and why their data more support that TYMS is an indicator of poor prognosis in ESCC patients?

4. The authors indicate the results in Figures 1C-E as an unpublished parallel multigroup study (line 272). Are these results their own data? If so, please provide more information in the methods section and it would be better to indicate these are their data because “an unpublished parallel multigroup study” is not a precise description.

5. The background and discussion sections are poorly written and lack focus.

6. There are many grammatical and typographical errors throughout the manuscript. For example, line 131 form ->from; lines 288, 292 knockout ->should be knockdown.

7. PLOS authors have the option to publish the peer review history of their article (what does this mean?). If published, this will include your full peer review and any attached files.

Reviewer #2: **Yes: **Shu-Ping Wang

---

## [Author Response · Author response to Decision Letter 1]

28 Jul 2023

Dear Editor

We want to thank you for your useful comments and suggestions on our manuscript. We have revised the manuscript accordingly, and detailed corrections are listed below:

Journal Requirements:

Author Response: 

Thank you for your suggestions. We have reviewed the reference, and ensure that it is complete, correct, and meet PLOS ONE's style requirements.

Comments of Reviewer 

1). This part is still a big confusion to the reviewer and maybe to the readers as well. Although the authors found out other reports to support their findings “TYMS promotes ESCC cell growth by relieving oxidative stress through activating Nrf2 expression” (e.g., Kimura et al. ESCC patients’ outcome [ref. 22], NSCLC report [ref.24]), the discrepancy in the relationship between the expression levels of TYMS and the outcome of cancer patients receiving chemotherapy/radiotherapy or not remains to be not addressed or discussed (Figure 1). Actually, although TYMS is highly expressed in ESCC tissue samples in the GEPIA data, the patients with high TYMS expression exhibit a better prognosis (Figures 1A and 1B). The similar trend can be found in Figure 1E, while these patients have received radiotherapy and chemotherapy. In contrast, the survival curve in patients without radiotherapy and chemotherapy indicates that the patients with high TYMS expression exhibit a worse prognosis (Figure 1D), which favors to the authors’ present study and experimental setting, that is “TYMS overexpression promotes ESCC cell growth in the condition without chemotherapy and/or radiotherapy”.

Author Response: 

Thanks a lot for your comments. We are so sorry for the confusion. Indeed, the complex results of survival analysis greatly confuse the reviewer and readers. The survival analysis results were easily disturbed by multiple factors (such as chemoradiotherapy). In Ref.22 and 24, none of ESCC and NSCLC patients included in the analysis received chemotherapy or radiation therapy prior to or following surgery, and results based on this were reliable: High TYMS mRNA expression levels had a significantly shorter survival after surgery. Ref.23 also confirmed the above point, and high tumor TYMS levels were associated with improved OS and disease-free survival (DFS) following an adjuvant therapy for colon cancer. However, we often overlook this point, and may get a contradictory result. After careful consideration, we decided to delete the survival analysis results to avoid this confusion from complex results, and only Figure 1A was reserved. The deleted results done not affect the conclusion of present study, and the viewpoint (ESCC patients with high TYMS expression exhibit a worse prognosis) was also supported by Ref.22. Thanks again for your comments and suggestion.

2). The authors have to clearly distinguish the clinical sample data with or without chemotherapy/radiotherapy because these two conditions seem to lead to opposite patients’ outcome. The current description for Figure 1 and the second paragraph in the discussion section kind of mix up these two conditions (i.e. with or without chemotherapy/radiotherapy) which is confusing and does not provide any more new insights.

Author Response:

Thanks a lot for your comments. I am so sorry again for the confusion from the survival analysis results. The detailed explanation could be found in Author Response 1. Thanks again for your suggestion.

3). Since chemotherapy and radiotherapy cause an increase in intracellular ROS that can lead to apoptosis via extrinsic or intrinsic pathways and based on the authors’ present study that TYMS activates Nrf2 expression and the following ROS scavenger events, it would be better to discuss the possibility why high TYMS expressing patients receiving chemotherapy and radiotherapy may have a better clinical outcome. Instead, the current discussion for the reports of the clinical outcome of patients who receive chemotherapy and radiotherapy or not would be better to move to the description for Figure 1 and mention that it seems there are two clinical conditions for the roles of TYMS in patients' survival and the author will focus on studying the condition without chemotherapy and radiotherapy.

Author Response:

Thanks a lot for your suggestions. The present study did showed that the high TYMS expressing patients receiving chemotherapy and radiotherapy may have a better clinical outcome. After a large number of investigation on the literatures, we still unable to reasonably explain this result, and it may be resulted from multiple factors. Therefore, it is a interesting clinical topics, and will be our future research focus. We also hope to expand the sample size verification to further confirm the relationship between the TYMS expression and radiotherapy, chemotherapy, prognosis to provide theoretical basis and molecular basis of TYMS for clinical prognostic indicators and drug use. Thanks again for your comments.

4). Are the patients in the GEPIA data receiving chemotherapy/radiotherapy or not and why their data more support that TYMS is an indicator of poor prognosis in ESCC patients?

Author Response:

Thanks for your comments. We do not get the accurate clinical information of patients who chemotherapy/radiotherapy or not. So the results of survival analysis from GEPIA was unreliable, and the results were finally deleted. Thanks again.

5).The authors indicate the results in Figures 1C-E as an unpublished parallel multigroup study (line 272). Are these results their own data? If so, please provide more information in the methods section and it would be better to indicate these are their data because “an unpublished parallel multigroup study” is not a precise description.

Author Response:

Thanks a lot for your comments. These results in Figures 1C-E was from unpublished parallel multigroup study (Transcriptome Sequencing), and the relevant results could not published or be found in any database. We also could not provide more information due to data confidentiality reasons. Therefore, we decided to delete these results. It could not affect the conclusion of present study, and the viewpoint (ESCC patients with high TYMS expression) was also supported by Ref.22 and GEPIA data. Thanks again for your suggestions.

6). The background and discussion sections are poorly written and lack focus.

Author Response:

Thanks for your comments. We have revised the background and discussion sections, and detailed corrections could be found in new manuscript.

7). There are many grammatical and typographical errors throughout the manuscript. For example, line 131 form ->from; lines 288, 292 knockout ->should be knockdown.

Author Response:

Thanks for your suggestion. We are so sorry for the grammatical and typographical errors. We have revised whole paper, and carefully modified these errors. Thanks again for your helps.

---

## [Decision Letter · Decision Letter 2]

4 Aug 2023

Thymidylate synthase promotes esophageal squamous cell carcinoma growth by relieving oxidative stress through activating nuclear factor erythroid 2-related factor 2 expression

PONE-D-23-05091R2

Dear Dr. Yang,

We’re pleased to inform you that your manuscript has been judged scientifically suitable for publication. **However, I strongly encourage all authors to double-check your English and grammar throughout the manuscript. **The abstract in your R2 version was partly modified from R1. As a result and unfortunately it does not read well (e.g. have payed, In vitro and in vivo found, throughout a MTT assay, etc.). I also noticed many run-on sentences if you could polish and improve all of them. For example, in the abstract, "Stably TYMS-overexpression cells were established by lentivirus transduction, and were used....., and RNA sequencing was performed...." can be changed to - Stably TYMS-overexpression cells established by lentivirus transduction were used for the analysis of cell proliferation. RNA sequencing was performed to explore the possible carcinogenic mechanisms.

Your manuscript will be formally accepted for publication once it meets all outstanding technical requirements.

Kind regards,

Yoshiaki Tsuji

Academic Editor

PLOS ONE

Additional Editor Comments (optional): IMPORTANT

**I strongly encourage all authors to double-check your English and grammar throughout the manuscript. **The abstract in your R2 version was partly modified from R1. As a result and unfortunately it does not read well (e.g. have payed, In vitro and in vivo found, throughout a MTT assay, etc.). I also noticed many run-on sentences if you could polish and improve all of them. For example, in the abstract, "Stably TYMS-overexpression cells were established by lentivirus transduction, and were used....., and RNA sequencing was performed...." can be changed to - Stably TYMS-overexpression cells established by lentivirus transduction were used for the analysis of cell proliferation. RNA sequencing was performed to explore the possible carcinogenic mechanisms.

Please check and correct all of these issues before publication.

Reviewers' comments:

Reviewer's Responses to Questions

**Comments to the Author**

1. If the authors have adequately addressed your comments raised in a previous round of review and you feel that this manuscript is now acceptable for publication, you may indicate that here to bypass the “Comments to the Author” section, enter your conflict of interest statement in the “Confidential to Editor” section, and submit your "Accept" recommendation.

Reviewer #2: All comments have been addressed

2. Is the manuscript technically sound, and do the data support the conclusions?

Reviewer #2: Yes

3. Has the statistical analysis been performed appropriately and rigorously? 

Reviewer #2: Yes

4. Have the authors made all data underlying the findings in their manuscript fully available?

Reviewer #2: Yes

5. Is the manuscript presented in an intelligible fashion and written in standard English?

Reviewer #2: Yes

6. Review Comments to the Author

Reviewer #2: (No Response)

7. PLOS authors have the option to publish the peer review history of their article (what does this mean?). If published, this will include your full peer review and any attached files.

Reviewer #2: No

---

## [Editor Report · Acceptance letter]

30 Aug 2023

PONE-D-23-05091R2 

Thymidylate synthase promotes esophageal squamous cell carcinoma growth by relieving oxidative stress through activating nuclear factor erythroid 2-related factor 2 expression 

Dear Dr. Yang:

I'm pleased to inform you that your manuscript has been deemed suitable for publication in PLOS ONE. Congratulations! Your manuscript is now with our production department. 

Kind regards, 

on behalf of

Dr. Yoshiaki Tsuji 

Academic Editor

PLOS ONE